# Estimates of the Global Burden of COVID-19 and the Value of Broad and Equitable Access to COVID-19 Vaccines

**DOI:** 10.3390/vaccines10081320

**Published:** 2022-08-15

**Authors:** Eleanor Bell, Simon Brassel, Edward Oliver, Hannah Schirrmacher, Sofie Arnetorp, Katja Berg, Duncan Darroch-Thompson, Paula Pohja-Hutchison, Bruce Mungall, Stuart Carroll, Maarten Postma, Lotte Steuten

**Affiliations:** 1Office of Health Economics, London SW1E 6QT, UK; 2Health Economics & Payer Evidence, BioPharmaceuticals Medical, AstraZeneca, 431 50 Gothenburg, Sweden; 3Access Policy & Strategy, BioPharmaceuticals Business, AstraZeneca, Sydney, NSW 2113, Australia; 4International Market Access, Vaccines and Immune Therapies, BioPharmaceuticals Business, AstraZeneca, Shanghai 201203, China; 5Global Policy, BioPharmaceuticals Business, AstraZeneca, Cambridge CB2 0AA, UK; 6Asia Area Medical Director, Vaccines and Immune Therapies, BioPharmaceuticals Business, AstraZeneca, Singapore 038986, Singapore; 7Department of Pharmacy, Unit of PharmacoEpidemiology & PharmacoEconomics, University of Groningen, 9712 CP Groningen, The Netherlands

**Keywords:** COVID-19, vaccines, value, access, burden

## Abstract

The objectives of this research were to produce a macro-level overview of the global COVID-19 burden and estimate the value of access to COVID-19 vaccines. A targeted literature review collated evidence of the burden. Linear modelling and data analysis estimated the health and economic effects of COVID-19 vaccines delivered in 2021, and whether additional value could have been achieved with broader and more equitable access. By 1 December 2020, there had been an estimated 17 million excess deaths due to COVID-19. Low-income countries allocated more than 30% of their healthcare budgets to COVID-19, compared to 8% in high-income countries. All country income groups experienced gross domestic product (GDP) growth lower than predicted in 2020. If all 92 countries eligible for COVAX Advance Market Committee (AMC), access had reached 40% vaccination coverage in 2021, 120% more excess deaths would have been averted, equivalent to USD 5 billion (10^9^) in savings to healthcare systems. Every USD spent by advanced economies on vaccinations for less advanced economies averted USD 28 of economic losses in advanced economies and USD 29 in less advanced economies. The cost to high-income countries when not all countries are vaccinated far outweighs the cost of manufacturing and distributing vaccines globally.

## 1. Introduction

The COVID-19 pandemic has had many far-reaching effects on health, the economy and society, which continue to evolve and are often intertwined. There is already a rich and wide-ranging body of literature on the consequences of COVID-19, but this is highly disparate in terms of outcomes considered, geographic scope, and methodological approach. Challenges related to data availability in some low- and middle-income countries (LMICs) also mean that the burden of the pandemic in these countries has often been underestimated [1,2,3].

Vaccines for COVID-19 have been developed at an unprecedented speed [4], and individuals in some countries have received four doses [5,6]. Yet as of 19 March 2022, only 14% of people in low-income countries had received at least one dose of a COVID-19 vaccine, compared to 79% of people in high-income countries [7]. There are compelling arguments to suggest that broad and equitable access to vaccines is not only a humanitarian imperative but is in everyone’s economic interest—global economies are highly interconnected through trade linkages, ongoing pandemic-related disruption in one country can affect the economic performance of other countries [8,9]—as well as being our best line of defence against new and emerging variants of SARS-CoV-2 [10,11]. 

This study had two objectives. Firstly, to provide an international perspective on the health and economic burden of COVID-19, by summarising the current literature on the magnitude and distribution of these effects worldwide. Secondly, to estimate the health and economic value generated by COVID-19 vaccines as of 1 December 2021 and the potential additional value generated by broader and more equitable access to vaccines. 

## 2. Materials and Methods

### 2.1. Estimating the COVID-19 Burden

This search aimed to identify geographically representative evidence relating to four elements of the COVID-19 burden of particular significance to patients, healthcare systems, and society overall. We considered effects on mortality using the indicators of estimated official deaths and excess deaths, because we recognise that official death counts provided may be limited by countries’ testing capacity and that COVID-19 can indirectly cause deaths among non-infected individuals, which makes excess deaths a more comprehensive—if uncertain—estimate of the total impact of the pandemic on deaths than the confirmed COVID-19 death count alone [12]. We also considered effects on quality of life (QoL), the second dimension through which COVID-19 and other infections can impact on the health of individuals. To consider QoL, we used the indicator of estimated quality-adjusted life years (QALYs) due to COVID-19. The quality-adjusted life year or QALY is a generic measure of disease burden, including both the quality and the quantity of life lived. 1 QALY is equal to 1 year in perfect health. QALYs are routinely used as a summary measure of health outcomes in economic evaluation, and thus are useful to contextualize the scale of COVID-19’s effects on health [13]. However, QALY estimates and other measures of the health effects of COVID-19 are not yet available for most countries, and so we complemented this indicator with the number of COVID-19 infections to provide insight into the scale of the expected health effects in different countries and country income groups worldwide. To estimate effects on healthcare system functioning, we used the indicator of the percentage of total per capita healthcare budgets which have been allocated to COVID-19. This indicator provides insight into how far the burden of COVID-19 has disrupted existing health system functioning, and allowed us to make use of robust cross-country data collected by the WHO [14]. To estimate effects on the macroeconomy, we used the indicator of changes in GDP, the ‘gold standard’ measure of an economy’s size and overall performance [15]. We complemented this with the indicator of changes in working hours due to COVID-19 to provide insight into how affected macroeconomic performance have impacted the livelihoods of populations. Definitions of these elements are provided in Appendix A. We searched PubMed, Scopus, Web of Science, and EconLit for English-language papers published up to 31 August 2021 using the search strategy provided in Appendix A. This search was complemented with an additional snowballing search strategy (using the reference lists of identified papers to identify further papers and grey literature, from Greenhalgh and Peacock 2005). We then updated any quantitative results which were derived from live datasets on 1 December 2021. 

For each value element (except effects on the macroeconomy, where the endogeneity between direct and indirect effects prevents identification of how these effects contributed to the total burden), we defined COVID-19’s effects on health and the economy as either direct or indirect. Direct effects of COVID-19 are produced by a COVID-19 infection, for example, a death caused by an infection or a loss of productivity because a patient with a COVID-19 infection cannot go to work. All effects which cannot be directly attributed to an individual’s infection with COVID-19 but are instead consequences of the circulation of COVID-19 in the population, and policy measures to address this, such as the QoL effects of mental health conditions created or exacerbated by lockdowns, are indirect [16]. Excess deaths measure the number of observed deaths compared to what would have been expected in the same time frame without the pandemic. It captures not only the confirmed deaths (reported direct deaths), but also COVID-19 deaths that were not correctly diagnosed and reported (unreported direct deaths) as well as deaths from other causes that are attributable to the overall crisis conditions (indirect deaths) [12].

The results are presented in a narrative synthesis structured by element, which describes the evidence identified for each outcome, and the distribution of these outcomes between lower- and higher-income countries. The results are disaggregated by country income groups, according to the World Bank’s classification [17], which categorises countries discretely as either high-income (HICs), upper-middle-income countries (upper MICs), lower-middle-income countries (lower MICs) or low-income countries (LICs), based on their gross national income per capita. 

Additionally, a heatmap/matrix of the identified evidence is used to demonstrate the availability of evidence, by geographic region, for the four elements. The colour-coding is used to demonstrate its credibility based on the quality of the indicator of the burden and its source. 

### 2.2. Estimating the Value of Global Access to COVID-19 Vaccines

#### 2.2.1. The Value of Global Access to COVID-19 Vaccines to Health and Healthcare Systems

A linear model estimated the value of vaccines to health and healthcare systems in 2021 for two scenarios: **Scenario 1** estimated the global value of vaccines in reducing direct (reported) and excess (estimated direct unreported and estimated indirect) deaths, direct hospitalisations, and direct healthcare system resource use due to COVID-19 in 2021, given the computed, country-specific monthly coverage rates between January and September 2021 [18]. Publicly available projections for country-specific vaccination rates for the time between September 2021 and December 2021 were applied to yield coverage estimates for the rest of the year [18].**Scenario 2** estimated the potential additional value that could have been achieved by the end of 2021 if at least 40% of the population in each of the 92 countries eligible for COVAX AMC (the AMC92) access would have been fully vaccinated. Scenario 2 is aligned with the World Health Organization (WHO) Strategy to Achieve Global COVID-19 Vaccination by mid-2022 [19] and the International Monetary Fund (IMF) Proposal to End the Pandemic [20]. The target was to vaccinate at least 40% in each country worldwide by the end of 2021, and the proposal was estimated to cost approximately USD 9 trillion, mainly in additional upfront grants to COVAX for purchasing vaccines [20]. 

Full details of our model are available in Appendix B.

#### 2.2.2. The Economic Value of Global Access to COVID-19 Vaccines 

To estimate the macroeconomic value of vaccines, we combined estimates from the International Chamber of Commerce (ICC) and IMF of the value and costs of global access to COVID-19 vaccines. **Estimates from the ICC:** The ICC’s paper “The Economic Case for Global Vaccinations” [21] uses an epidemiological susceptible−infected−recovered (SIR) model with an international trade and production network to estimate the economic costs of COVID-19 during a single year that were solely due to international linkages. The most realistic scenario (as identified by the authors) assumed that, in advanced economies, half of the susceptible population was vaccinated within the first 30 days of the vaccination programme starting, and the remaining half was vaccinated within the following 90 days. In less advanced economies, the vaccination program started at the same time, but it took a full year to vaccinate half of the susceptible population. The model estimated the economic losses associated with this scenario, compared to a hypothetical scenario of full vaccination in both advanced economies and less advanced economies. In other words, this scenario provides an estimate of the economic value which would be generated if vaccination coverage in less advanced economies increased from 50% to 100%. Note that the IMF categorises all countries as advanced economies, emerging markets, or developing economies. The main criteria used are [1] per capita income level, [2] export diversification—so oil exporters that have high per capita GDP would not make the advanced classification because around 70% of their exports are oil—and [3] degree of in-tegration into the global financial system. For the purposes of this article, we refer to emerging markets and developing countries collectively as ‘less advanced economies’ [22].**Estimates from the IMF:** The IMF’s Proposal to End the Pandemic [20], launched in May 2021, set the target of vaccinating at least 40% of the population of every country worldwide by the end of 2021 and 60% by mid-2022, and provided estimates of the costs of this proposal. To generate a conservative estimate of the total incremental costs of fully vaccinating every country, we summed the cost of vaccinating 60% of the AMC92 countries (USD 50 billion for the IMF’s proposal to reach 40% vaccination coverage in every country worldwide by the end of 2021 and 60% by mid-2022) with the costs of increasing coverage from 60% to 100% in lower MICs (USD 16 billion, assuming a cost of USD 4 billion per 10 percentage point increase, as indicated in the IMF’s proposal). This assumes that HICs have already purchased or ordered vaccines sufficient to achieve at least 100% domestic coverage. **Analysis:** Dividing the economic value of moving from 50% to 100% vaccination coverage in less advanced economies, by the cost of moving from current vaccination levels (in May 2021, when the IMF proposal was written, and most countries eligible for AMC92 access had not reached 20% vaccination coverage) to 100% vaccination, provides a conservative estimate of the return on investment to vaccination. Full details of how we used these sources to develop estimates of the economic value of global access to vaccines are available in Appendix C. 

## 3. Results

### 3.1. Estimates of the COVID-19 Burden

#### 3.1.1. Impact on Length of Life

Highly credible (i.e., from a highly credible source, either peer-reviewed or key data repositories) evidence on the direct mortality burden of COVID-19 is available across all regions, but an evidence gap exists regarding the indirect burden of COVID-19 in South Asia and sub-Saharan Africa (see Figure 1). Direct fatalities stemming from the pandemic may be underreported in these regions, which are predominantly comprised of LICs and lower MICs [17].

The number of official COVID-19 deaths reported globally and by country income group is presented in Figure 2 (per million population) and Table 1 (absolute figures). Due to differences in definitions, reporting, and testing facilities, it is likely that these results underestimate the number of deaths directly due to COVID-19 in some countries [12]. Furthermore, the pandemic has indirectly caused deaths by disrupting the provision of and access to healthcare. 

As Table 1 shows, rates of official deaths are highest in upper MICs and HICs. However, the much higher ratios of excess deaths to official deaths in lower MICs and particularly LICs indicates that official deaths have been significantly underestimated in these country income groups.

#### 3.1.2. Impact on QoL

Figure 1 shows that there is a lack of high-quality evidence on COVID-19′s direct and indirect effects on QoL worldwide, particularly in the sub-Saharan Africa region. Only one study to date, which used data from the UK, has estimated the number of QALYs lost due (directly and indirectly) to COVID-19 between March 2020 and February 2021 [23]. A total of 81,000 QALYs were lost due to morbidity from COVID-19 in the UK during this period. Far more data is available on COVID-19 infections. The number of COVID-19 infections reported globally and by country income group is shown in Figure 3 (per million population) and Table 2 (absolute figures). As for effects on length of life, differences in definitions, testing, and reporting mean that these results likely underestimate the number of infections in some countries [12], and the true number of cases is estimated to be far higher. Infection rates are highest in HICs, followed by upper MICs and lower MICs. The infection rates observed in HICs are more than 100 times higher than the rates observed in LICs. However, this is likely to be partly explained by correlation between countries’ income and capacity for reporting and testing [12]. More research is needed to understand how COVID-19 infections affect QoL, and in particular the full consequences of long COVID, which are as yet unknown. 

There are at present no available estimates of COVID-19′s indirect effects on QoL, nor of proxy indicators, at the global or country income group level. 

#### 3.1.3. Impact on Health System Resource Use

As Figure 1 shows, there are significant evidence gaps across all regions in estimates of COVID-19′s direct and indirect effects on health system resource use. 

Table 3 shows per capita budget allocations for the COVID-19 response in 2020 by country income group, in absolute terms and as a proportion of per capita government spending on health in 2018, for a sample of 16 LICs, 60 upper and lower MICs, and 37 HICs (data are available only for lower MICs and upper MICs collectively). This provides a proxy indicator of the direct impact of COVID-19 on health system resource use. Even though low-income countries allocated far less to the COVID-19 response than middle- and high-income countries, this equates to a much larger proportion of their overall health spending, which indicates that the COVID-19 burden is larger relative to health system capacity in low-income countries. 

In addition to the direct impact of COVID-19 on health system resource use, there will also be indirect consequences. Delays in diagnosis and treatment during the pandemic, when resources have been diverted to COVID-19, are expected to lead to permanent deteriorations in patients’ conditions that will not only have negative effects on patients’ lives but make those conditions more expensive to treat [24]. However, there is very little evidence available on these indirect consequences, the full effects of which will not be felt for several years.

#### 3.1.4. Impact on Macroeconomic Performance

Macroeconomic effects can be produced directly by reducing the productivity of workers infected with COVID-19, and indirectly through lockdowns and other pandemic-related restrictions. However, it is not possible using available data to disaggregate overall effects on GDP into those attributable to direct and indirect causes. There are credible proxy measures of COVID-19′s overall impact on macroeconomic performance available in every country, which exist in the form of deviation from the pre-COVID-19 projections of GDP (see Figure 1). Table 4 summarises these results by country income group and shows that the difference between project and estimated GDP growth in 2020 were largest in upper MICs, followed by HICs, lower MICs, and LICs. 

One channel through which COVID-19 has affected macroeconomic performance is reduction in working hours and employment. Across the world, people’s ability to work during the pandemic has been hindered by government restrictions on workplaces, supply disruption, and macroeconomic contraction in general. Table 5 shows the percentage change in working hours from 2020 to 2019 by country income group. The decline was largest in lower MICs, followed by HICs, upper MICs, and LICs.

### 3.2. The Value of COVID-19 Vaccines

#### 3.2.1. The Value of COVID-19 Vaccines to Health and Healthcare Systems

Our model’s results under **Scenario 1** show that in 2021 1.4 million direct deaths, 4.3 million excess (direct and indirect) deaths and 6.0 million hospitalisations have been averted and hospital resources worth USD 59 billion have been saved by COVID-19 vaccinations. Results for **Scenario 2** show that, if the 40% vaccination target had been reached, a total of 1.8 million direct deaths, 9.5 million excess deaths, and 7.6 million hospitalisations of COVID-19 patients could have been averted, and hospital resources worth USD 64 billion have been saved. This is equivalent to more than twice the number of excess deaths averted under **Scenario 1**, and more than 25% more direct deaths and potential hospitalisations averted. The smaller increase in hospital resource use reflects relatively lower healthcare resource use costs in countries eligible for COVAX AMC access. Our results state that hospitalisations and hospital resource use could have been potentially averted, as opposed to averted. This is because the Institute for Health Metrics and Evaluation (IHME) data used in our modelling predict the total need for hospital care, but do not consider healthcare system capacity constraints for delivering this care [25]. 

Figure 4 provides estimates of the value of vaccination in avoiding direct deaths, excess deaths, and hospitalisations under **Scenario 1** and **Scenario 2**.

Table 6 shows our estimates of the value of vaccines for these outcomes, as well as effects on the number of hospital beds, ICU beds, and the value of these effects on bed usage to healthcare systems under Scenario 1 and Scenario 2, including ranges based on +/− 5 percentage point input values for vaccine efficacy.

Figure 5 shows the value of vaccination by country income group in terms of the number of indirect and excess deaths avoided per 10,000 population. If the 40% target had been reached, the number of deaths avoided per 100,000 people would have increased from 68 to 227 in lower MICs and from 38 to 536 in LICs. 

#### 3.2.2. The Economic Value of COVID-19 Vaccines 

By increasing the size of the healthy workforce and reducing the need for lockdowns and other pandemic-related restrictions, vaccines are key to a country’s recovery from the macroeconomic burden caused by COVID-19 [26]. However, because global economies are highly interconnected through trade linkages, ongoing pandemic-related disruption in one country can affect the economic performance of other countries. To date, the most extensive macroeconomic model estimating the costs of delays to vaccination suggests a global loss of USD 3.8 trillion over one year if less advanced economies reached 50% vaccination as opposed to a hypothetical 100% vaccination (the model assumes 100% vaccination and therefore full economic recovery in advanced economies; the authors used the IMF’s categorisation of countries as advanced economies, emerging market, and middle income economies, and low-income developing countries.). These results are conservative, given that many less advanced economies did not reach this threshold, and translate to a loss of more than 4% of pre-pandemic global GDP [21,27]. Of the USD 3.8 trillion loss, the model estimates that 49% (USD 1.86 trillion) would fall on advanced economies. This is due to their trade linkages with less advanced economies, which would bear the remaining 51% (USD 1.91 trillion). 

The analysis below estimates that it will cost USD 66 billion to reach 100% vaccine coverage in the 92 countries eligible for AMC access. This is consistent with the figures published by the Access to COVID-19 Tools (ACT) Accelerator [28]. By dividing the expected macroeconomic benefits to advanced economies of increasing global vaccine coverage from 50% to 100% (USD 1.86 trillion) by this cost (USD 66 billion), we generated a conservative estimate of the return on investment for HICs if they were to finance 100% of the vaccination coverage in the AMC92. As Figure 6 illustrates, our estimates suggest that, for every 1 USD spent by advanced economies, they could expect to avert 28 USD of economic losses in a single year. A further 29 USD of economic losses would also be averted in less advanced economies (USD 1.91 trillion ÷ USD 66 billion). The collective cost to HICs when not all countries are vaccinated therefore heavily outweighs the cost of manufacturing and distributing vaccines globally. 

The size of the economic losses averted in a specific country depend on the sectoral composition of its economy. In HICs, the sectors which bear the highest economic costs are those that are most exposed to trade with less advanced economies, such as textiles and apparel, basic metals, and food and beverages [21]. Therefore, countries whose economy depends to a large extent on these sectors will be most affected by the continuing disruption caused by the pandemic and accrue the greatest benefits from investing in global access to vaccination. 

## 4. Discussion

### 4.1. Estimates of the COVID-19 Burden

This paper collates the evidence of the global burden of COVID-19 for four key outcomes: length of life, QoL healthcare system resource use, and macroeconomic performance. While credible estimates of COVID-19′s effects on length of life and macroeconomic performance are available in every region worldwide, there are gaps in evidence for COVID-19′s indirect effects on QoL in sub-Saharan Africa, and major gaps worldwide in evidence for COVID-19′s effects on healthcare system resource use. This paper also summarises the best available evidence on the same four outcomes and provides evidence that countries in all income groups have been affected by the pandemic across all of these dimensions. While infection and mortality rates have been highest in HICs and upper MICs, there is also evidence that official death rates in LICs and lower MICs significantly underestimate the true burden in these countries. Governments in LICs have also allocated a much greater proportion of their healthcare budgets to the COVID-19 response than those in upper and lower MICs and HICs, which is likely to have major repercussions for the delivery of other healthcare and therefore the indirect health impact of COVID-19. Moreover, the absolute amounts spent by LICs per capita are absolutely much smaller than in other countries, meaning that they are worst affected in terms of access to vaccines. COVID-19′s impact on macroeconomic performance in 2020 was largest in HICs and upper MICs, although there is emerging evidence that these countries are also recovering faster [29].

### 4.2. Estimates of the Health Value of COVID-19 Vaccines

Our estimates of the health value of vaccines demonstrate that COVID-19 vaccines had substantial positive effects on deaths, hospitalisations, and healthcare system resource use in 2021, but that these effects could have been increased—by more than 50% in the case of excess deaths—if all 92 countries eligible to receive COVAX AMC access had achieved a 40% vaccination rate in their population by the end of 2021, as per the WHO and IMF targets [19,20]. 

### 4.3. Estimates of the Economic Value of COVID-19 Vaccines

Our estimates of the economic value of vaccines rely on the model produced by Çakmaklı et al. [21]. A full description of the limitations is available in the paper but include the limited sample of 65 countries to which the model is calibrated and the assumption of static global value chains, where producers and suppliers do not optimise as a response to the pandemic shock (by therefore restricting the scope to the first-round effects of the pandemic, this produces a conservative estimate of macroeconomic disruption and therefore of the economic value of vaccines).

### 4.4. Limitations

Our estimates of the COVID-19 burden are limited by gaps in the available evidence, along with the uncertainty associated with COVID-19′s long-term effects. The long-term QoL effects of ‘long COVID’ and disruption to the provision of other healthcare during the pandemic are still unknown, and there are also very limited data available on short-term QoL effects, while the scale of COVID-19′s macroeconomic effects are challenging to discern when many COVID-19-related unemployment and financial support programmes are still ongoing. These limitations suggest that our results are likely to be conservative. 

The main limitation of our modelling to estimate the health value of vaccines is that it likely significantly underestimates the value of vaccinations for preventing death and hospitalisation—although in doing so it provides a conservative estimate from which to calculate the relative value of broader and more equitable access to vaccines. The most important reason for the underestimation is that these outcomes are mainly observed in older age groups and vulnerable populations. We, however, calculated coverage rates across the total population based on fully vaccinated individuals independent of their age or risk status. As most governments prioritised the vaccination of people of higher age or higher vulnerability, our model leads to significantly lower estimates compared to models that applied age-group-specific coverage rates. In addition, we only considered fully vaccinated individuals. This is conservative as it ignores the value of administered first shots within a 2-dose schedule in the primary vaccine series. Finally, the estimates for the indirect death to direct deaths ratios may be biased by measurement error and may have a larger impact in LICs due to the small numbers of observed deaths. 

In addition, there are two potential key sources of additional value which our model does not attempt to capture: the value of vaccines in reducing transmission and mutation to new variants. In reducing transmission, COVID-19 vaccines help to reduce the total number of infections (among the vaccinated and unvaccinated), and consequently the associated mortality, morbidity, and healthcare system resource use. More complex, dynamic transmission models would be needed to quantify these additional benefits. By offering the virus fewer opportunities to mutate, COVID-19 vaccines may reduce the risk of new variants emerging [10,11]. Not only this, but by ‘priming’ the immune system, they can also reduce the severity of disease experienced by vaccinated patients infected with new variants, as emerging evidence on the Omicron variant suggests [30,31]. However, there is also some evidence to suggest that the Omicron variant may exhibit immune escape, which could in theory increase the probability of new variants [32]. 

Our estimates of the economic value of vaccines rely on cost estimates produced by the IMF, which are subject to high levels of uncertainty. However, the scale of the return on investment in vaccines to advanced economies which we estimate (USD 28 of benefit, to every USD 1 invested in vaccines) provides confidence in our key result, which is that it is in the economic interest of advanced economies to invest in global access to vaccines. 

## 5. Conclusions

In conclusion, we have demonstrated that COVID-19 has placed a substantial burden on health, health systems, and the macroeconomy of countries worldwide. Although healthcare systems in LICs have allocated the highest proportion of their budgets to COVID-19, the absolute levels of this spending are much lower than in other countries, suggesting that additional investment in COVAX or similar programmes would be required to facilitate access to vaccines in these countries. We have also demonstrated that it is in the self-interest of richer countries to invest in this access. Broader and more equitable access COVID-19 vaccines would not only reduce the health burden in lower-income countries, but simultaneously provide global macroeconomic benefits that far outweigh their cost. Advanced economies are conservatively expected to make a 28-fold return on investment in providing vaccine access to less advanced economies. While the true value of vaccination is impossible to fully quantify, the COVID-19 pandemic has shown the true scale of its benefits and impact. Broader and more equitable access to COVID-19 vaccines is not only a humanitarian imperative, but in the world’s economic interest. Finally, we note that as data about the pandemic increasingly become available, there will be an unprecedented opportunity for researchers to quantify many of the broader dimensions of the value of vaccines, which have historically been challenging to measure. Such research has a vital role to play in demonstrating the importance of investment in vaccine research and development and pandemic preparedness, and thereby reducing the risk posed by pandemics in the future.

## Figures and Tables

**Figure 1 vaccines-10-01320-f001:**
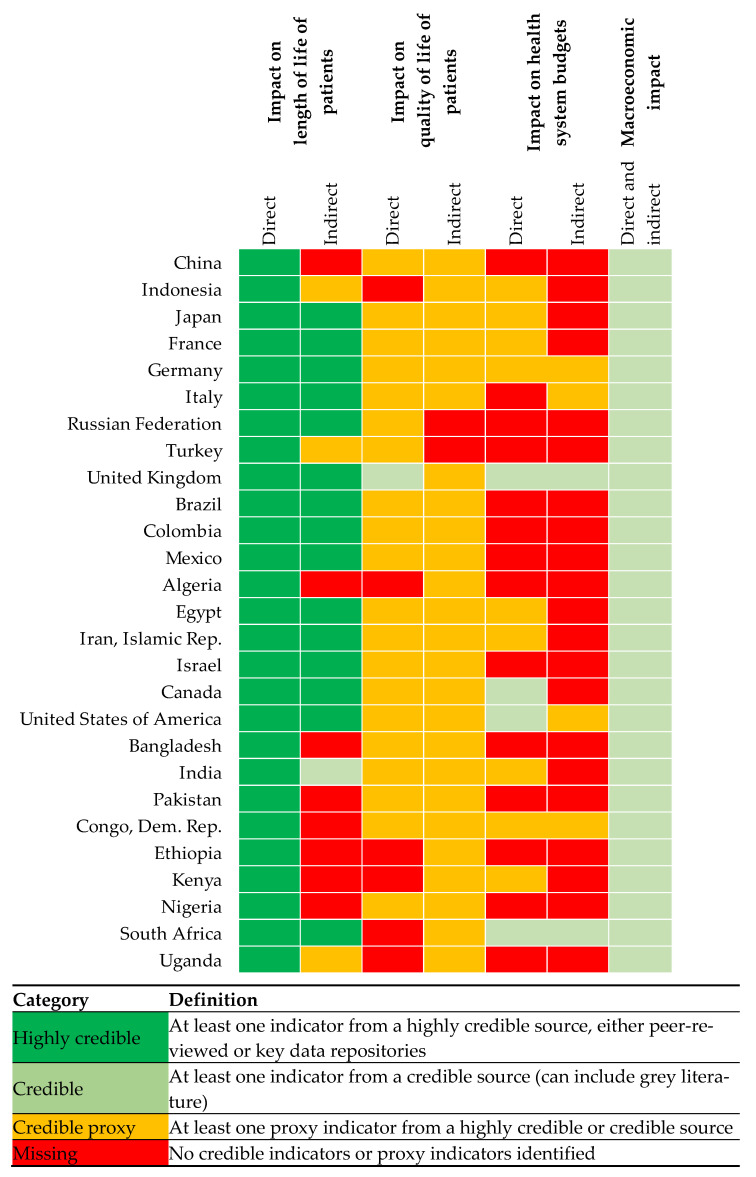
Heatmap of available evidence by value element and region, colour-coded by level of credibility.

**Figure 2 vaccines-10-01320-f002:**
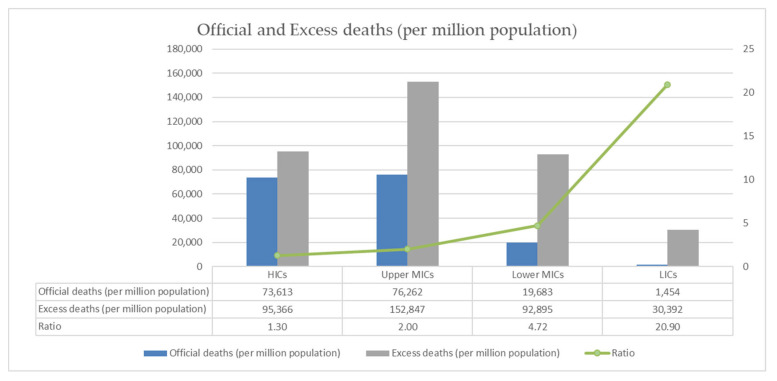
Global estimates of the mortality burden of COVID-19 as of 1 December 2021.

**Figure 3 vaccines-10-01320-f003:**
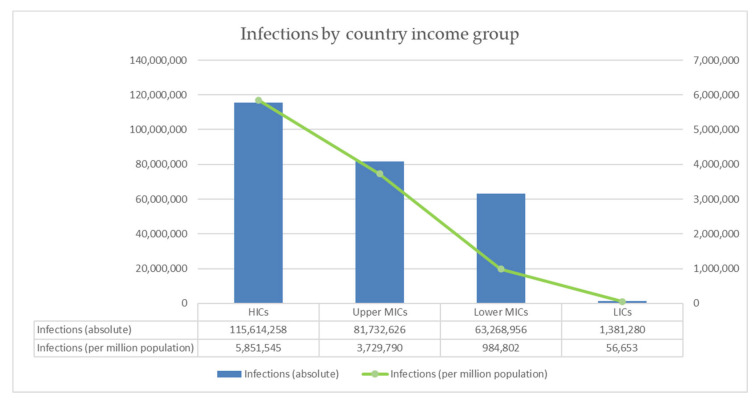
Global estimates of global SARS-CoV-2 infections as of 1 December 2021, derived from Ritchie et al. 2020.

**Figure 4 vaccines-10-01320-f004:**
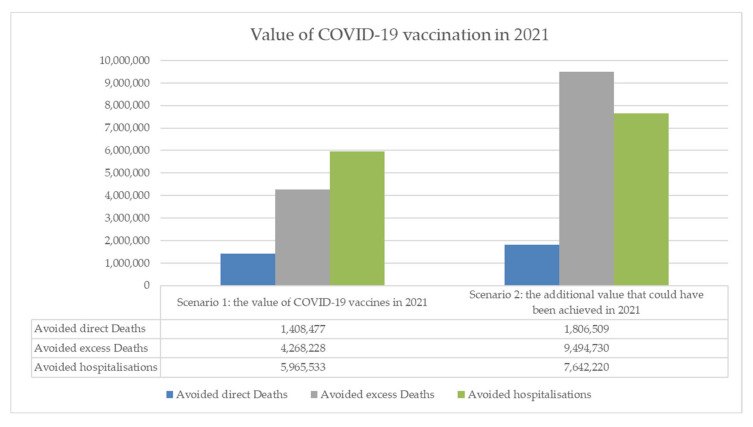
The value of vaccines in avoiding direct deaths, excess deaths, and hospitalisations in 2021.

**Figure 5 vaccines-10-01320-f005:**
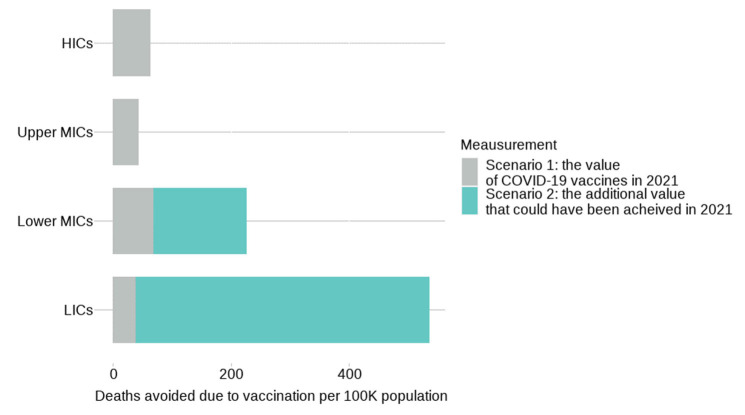
Excess deaths avoided due to vaccination per 100 k population per country income group.

**Figure 6 vaccines-10-01320-f006:**
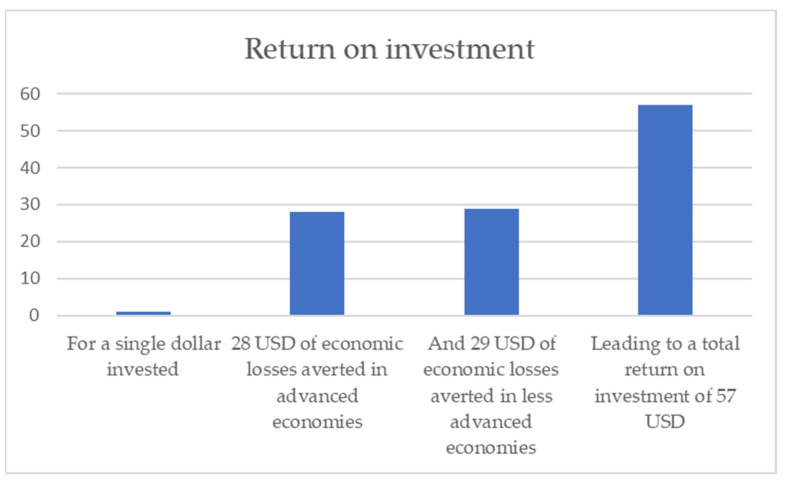
The return on investment to spending on broader and more equitable access to COVID-19 vaccines.

**Table 1 vaccines-10-01320-t001:** Global estimates of the mortality burden of COVID-19 as of 1 December 2021.

Indicator	Global	HICs	Upper MICs	Lower MICs	LICs
Official deaths	5,200,935	1,834,372	2,178,142	1,151,044	37,377
Excess deaths	17,656,843	2,285,593	5,166,016	9,230,371	974,862

**Table 2 vaccines-10-01320-t002:** Global estimates of global SARS-CoV-2 infections as of 1 December 2021, derived from Ritchie et al. 2020.

Indicator	Global	HICs	Upper MICs	Lower MICs	LICs
Number of infections	261,997,120	115,614,258	81,732,626	63,268,956	1,381,280

**Table 3 vaccines-10-01320-t003:** Spending on COVID-19 as a proportion of health system budgets [14] ^1^.

Indicator	HICs	Upper MICs and Lower MICs	LICs
Per capita budget allocations for the COVID-19 response in 2020	USD 205	USD 20	USD 3.20
Per capita government spending on health, 2018	USD 2519	USD 158	USD 8.9
Percentage of government spending allocated to COVID-19	8.1%	12.7%	36.4%

^1^ Data on 16 low-income, 60 middle-income, and 37 high-income countries compiled from a range of sources by the WHO. Full details of the country classification methodology are available from the WHO 2020.

**Table 4 vaccines-10-01320-t004:** Projected and estimated GDP growth in 2020 (IMF 2021b).

Indicator	HICs	Upper MICs	Lower MICs	LICs
Pre-COVID-19 projection	2.062457627	4.552076923	4.072865385	4.71096
Revised estimate	−6.736576271	−7.321538462	−2.835826923	−0.80292
Difference	8.799033898	11.87361538	6.908692308	5.51388

**Table 5 vaccines-10-01320-t005:** Percentage change in working hours in 2020 compared to 2020.

Indicator	HICs	Upper MICs	Lower MICs	LICs
Percentage change	−8.3%	−7.3%	−11.3%	−6.7%

**Table 6 vaccines-10-01320-t006:** Estimates of the value of vaccines including ranges based on +/− 5 percentage point input values for vaccine efficacy.

Scenario 1	Avoided Direct Deaths	Avoided Excess Deaths	Avoided Hospitalisations	Avoided Hospital Beds	Avoided ICU Beds	Value of Beds Saved
Base Case	1,408,477	4,268,228	5,965,533	54,872,848	16,422,386	58,859,819,280
Lower Bound	1,274,216	3,922,598	5,394,578	49,631,923	14,869,634	52,908,343,609
Upper Bound	1,561,553	4,646,739	6,597,764	60,680,927	18,139,229	65,495,842,196

## Data Availability

Data used to estimate the COVID-19 burden is available online: https://ourworldindata.org/coronavirus (accessed on 2 December 2021). Data used in modelling to estimate the health value of vaccines is available online: http://www.healthdata.org/covid/faqs (accessed on 2 December 2021).

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
