# Peer review of "Estimates of the Global Burden of COVID-19 and the Value of Broad and Equitable Access to COVID-19 Vaccines"

_vaccines, 2022, doi:10.3390/vaccines10081320_

Round 1
Reviewer 1 Report
Overall, this is a well structured, well written and very informative paper that trying to highlighted crucial subjects of discussion that have beed arised during the covid 19 pandemic period.
General Questions:
1. Regarding the indicators that Authors used, please explain why you choose these indicators and add the appropriate references.
Minor Commnets:
Line 33: add full stop
Line 26: add in parenthesis what GPD means.
Line 93: add reference
Line 99, line 137, line 330, line 368: add reference with the appropriate reference style (16)
Major Comments:
1. It is common for the introduction to end up by stating the research aims. In this context the two research aims have to be more clearly defined at the end of the introduction. Particularly, line 49-50 and Line 60-62 : the objectives of the study must be well defined and noted separately after the section of the Introduction
2. Ref No 25 must be replaced. This is not an official webpage that we could trust the information and evidence that they are referred to.
3. Add a paragraph with the limitations of the study. Limitations are referred during the text but this is not the appropriate way to do so.
Author Response
General Questions:
- Regarding the indicators that Authors used, please explain why you choose these indicators and add the appropriate references.
We have added the following paragraph with references:
This search aimed to identify geographically representative evidence relating to four elements of the COVID-19 burden of particular significance to patients, healthcare systems, and society overall. We consider effects on mortality using the indicators of estimated official deaths and excess deaths, because we recognise that official death counts may provide be limited by countries’ testing capacity and that COVID-19 can indirectly cause deaths amongst non-infected individuals, which makes excess deaths a more comprehensive – if uncertain – estimate of the total impact of the pandemic on deaths than the confirmed COVID-19 death count alone 1. We also consider effects on quality of life (QoL), the second dimension through which COVID-19 and other infections can impact on the health of individuals. To consider QoL, we use the indicator or estimated Quality-Adjusted Life Years (QALYs)[1] due to COVID-19. QALYs are routinely used as a summary measure of health outcome in economic evaluation, and thus are useful to contextualize the scale of COVID-19s effects on health 2. However, QALY estimates and other measures of the health effects of COVID-19 are not yet available for most countries, and so we complement this indicator with the number of COVID-19 infections to provide insight into the scale of the expected health effects in different countries and country income groups worldwide. To estimate effects on healthcare system functioning, we use the indicator of the percentage of total per capita healthcare budgets which have been allocated to COVID-19. This indicator provides insight into how far the burden of COVID-19 has disrupted existing health system functioning, and allows us to make use of robust cross-country data collected by the WHO 3. To estimate effects on the macroeconomy, we use the indicator of changes in GDP, the ‘gold standard’ measure of an economy’s size and overall performance 4. We complement this with the indicator of changes in working hours due to COVID-19 to provide insight into how effects on macroeconomic performance have impact the livelihoods of populations.
(1) Ritchie, H.; Ortiz-Ospina, E.; Beltekian, D.; Mathieu, E.; Hasell, J.; Macdonald, B.; Giattino, C.; Appel, C.; Rodés-Guirao, L.; Roser, M. Coronavirus Pandemic (COVID-19). Our World in Data 2020.
(2) Whitehead, S. J.; Ali, S. Health Outcomes in Economic Evaluation: The QALY and Utilities. British Medical Bulletin 2010, 96 (1), 5–21. https://doi.org/10.1093/bmb/ldq033.
(3) WHO. Global Spending on Health 2020: Weathering the Storm; World Health Organzation: Geneva, 2020.
(4) Brinkman, R. L.; Brinkman, J. E. GDP as a Measure of Progress and Human Development: A Process of Conceptual Evolution. Journal of Economic Issues 2011, 45 (2), 447–456. https://doi.org/10.2753/JEI0021-3624450222.
Minor Comments:
Line 33: add full stop - added
Line 26: add in parenthesis what GPD means - added
Line 93: add reference - added
Line 99, line 137, line 330, line 368: add reference with the appropriate reference style (16) – added with appropriate reference style
Major Comments:
- It is common for the introduction to end up by stating the research aims. In this context the two research aims have to be more clearly defined at the end of the introduction. Particularly, line 49-50 and Line 60-62 : the objectives of the study must be well defined and noted separately after the section of the Introduction
- Ref No 25 must be replaced. This is not an official webpage that we could trust the information and evidence that they are referred to. – We have removed this reference. We now reference only the following source:
Save the Children. Rich countries need to spend just $0.80 a week per citizen to stock world with COVID-19 vaccines [Internet]. Save the Children International. 2021 [cited 2021 Nov 16]. Available from: https://www.savethechildren.net/news/rich-countries-need-spend-just-080-week-citizen-stock-world-covid-19-vaccines
- Add a paragraph with the limitations of the study. Limitations are referred during the text but this is not the appropriate way to do so. – We have added the following paragraph (comprising text which mainly previously appeared in other sections)
Our estimates of the COVID-19 burden are limited by gaps in the available evi-dence, along with the uncertainty associated with COVID-19’s long-term effects. The long-term QoL effects of ‘long COVID’ and disruption to the provision of other healthcare during the pandemic are still unknown, and there is also very limited data available on short-term QoL effects, whilst the scale of COVID-19’s macroeconomic effects are challenging to discern when many COVID-19 related unemployment and financial support programmes are still ongoing. These limitations suggest that our re-sults are likely to be conservative.
The main limitation of our modelling to estimate the health value of vaccines is that it likely significantly underestimates the value of vaccinations for preventing death and hospitalisation – although in doing so it provides a conservative estimate from which to calculate the relative value of broader and more equitable access to vaccines. The most important reason for the underestimation is that these outcomes are mainly observed in older age groups and vulnerable populations. We, however, calculated coverage rates across the total population based on fully vaccinated individuals independent of their age or risk status. As most governments prioritised the vaccination of people of higher age or higher vulnerability, our model leads to significantly lower estimates compared to models that applied age-group specific coverage rates. In addition, we only consider fully vaccinated individuals. This is conservative as it ignores the value of administered first shots within a 2-dose schedule in the primary vaccine series. Finally, the estimates for the indirect death to direct deaths ratios may be biased by measurement error and may have a larger impact in LICs due to the small numbers of observed deaths.
In addition, there are two potential key sources of additional value which our model does not attempt to capture: the value of vaccines in reducing transmission and mutation to new variants. In reducing transmission, COVID-19 vaccines help to reduce the total number of infections [amongst the vaccinated and unvaccinated], and conse-quently the associated mortality, morbidity, and healthcare system resource use. More complex, dynamic transmission models would be needed to quantify these additional benefits. By offering the virus fewer opportunities to mutate, COVID-19 vaccines may reduce the risk of new variants emerging (10,11). Not only this, by ‘priming’ the immune system, they can also reduce the severity of disease experienced by vaccinated patients infected with new variants, as emerging evidence on the Omicron variant suggests (30,31). However, there is also some evidence to suggest that the Omicron variant may exhibit immune escape, which could in theory increase the probability of new variants (32).
Our estimates of the economic value of vaccines rely on cost estimates produced by the IMF, which are subject to high levels of uncertainty. However, the scale of the return on investment in vaccines to advanced economies which we estimate [28 USD of bene-fit, to every 1 USD invested in vaccines] provides confidence in our key result, which is that it is in the economic interest of advanced economies to invest in global access to vaccines.
[1] The Quality-Adjusted Life Year or QALY is a generic measure of disease burden, including both the quality and the quantity of life lived. 1 QALY is equal to 1 year in perfect health.
Reviewer 2 Report
I enjoyed reading the article. It is well documented and can be very useful for people making decisions related to vaccination.
The following changes should be made. On the first page, the addresses of the authors are incomplete; only the institution appears, but neither the country nor the city nor the department.
Only the e-mail addresses of 2 of the authors appear. The authors have introduced a footnote in the abstract; this should be avoided because it makes indexing in databases more difficult. They should insert the text in Brackets
The abstract is very long. The instructions for authors of the vaccine journal indicate that the abstract should be less than 200 words. The abstract exceeds 440 words. It should be made much shorter.
The article's authors repeatedly use the acronym QoL, but nowhere is its meaning explained The first time an abbreviation is introduced in the manuscript, it should be explained.
There are differences in the use of the word billion between the United States and the rest of the world. In the United States, a billion is 1000 million, while in the rest of the world, a billion is 1 000 000 million. The American billion is called a Millard in Europe, and the European billion is called a trillion in the United States. This is so confusing that the first time the authors introduce the word billion, they must indicate if it is 10^6 or 10^12.
Authors must consistently use the bibliography. Most of the time, cite the bibliography with numbers between parenthesis (Vancouver system), but sometimes you cite it with the APA method, as in line 137. (Çakmaklı et al., 2021a]
Author Response
The following changes should be made. On the first page, the addresses of the authors are incomplete; only the institution appears, but neither the country nor the city nor the department.
- We have added the country, city, and department where relevant (the Office of Health Economics has no departments)
Only the e-mail addresses of 2 of the authors appear. The authors have introduced a footnote in the abstract; this should be avoided because it makes indexing in databases more difficult. They should insert the text in Brackets
- We have added all email addresses and deleted the footnote in the abstract.
The abstract is very long. The instructions for authors of the vaccine journal indicate that the abstract should be less than 200 words. The abstract exceeds 440 words. It should be made much shorter.
- We have shortened the abstract which is now 200 words
The article's authors repeatedly use the acronym QoL, but nowhere is its meaning explained The first time an abbreviation is introduced in the manuscript, it should be explained.
- We have added the explanation on line 74
There are differences in the use of the word billion between the United States and the rest of the world. In the United States, a billion is 1000 million, while in the rest of the world, a billion is 1 000 000 million. The American billion is called a Millard in Europe, and the European billion is called a trillion in the United States. This is so confusing that the first time the authors introduce the word billion, they must indicate if it is 10^6 or 10^12.
- We have indicated in the abstract that we refer to the American billion (10^9)
Authors must consistently use the bibliography. Most of the time, cite the bibliography with numbers between parenthesis (Vancouver system), but sometimes you cite it with the APA method, as in line 137. (Çakmaklı et al., 2021a]
- We have updated the bibliography
Round 2
Reviewer 1 Report
Overall, this is a clear, concise, and well-written manuscript; the introduction is relevant and theory based; the appropriate information regarding the study findings are presented properly and help readers to follow the results and conclusion.
The Authors took under consideration all the points that were been highlighted and they proceeded them properly.